# Comparative Ecotoxicological Effects of Cyanobacterial Crude Extracts on Native Tropical Cladocerans and *Daphnia magna*

**DOI:** 10.3390/toxins17060277

**Published:** 2025-06-02

**Authors:** Cesar Alejandro Zamora-Barrios, Marcos Efrén Fragoso Rodríguez, S. Nandini, S. S. S. Sarma

**Affiliations:** 1Laboratory of Water Pollutants Removal Processes, Division of Research and Postgraduate Studies, Universidad Nacional Autónoma de México, FES-Iztacala, Tlalnepantla 54090, State of Mexico, Mexico; fragoso0905@gmail.com; 2Laboratory of Aquatic Zoology, Division of Research and Postgraduate Studies, Universidad Nacional Autónoma de México, FES-Iztacala, Tlalnepantla 54090, State of Mexico, Mexico; sarma@unam.mx

**Keywords:** native species, crude extract, FCHABs, relative sensitivity, life table

## Abstract

Freshwater cyanobacterial harmful algal blooms (FCHABs) alter zooplankton communities, often adversely, through the production of cyanotoxins. While *Daphnia magna* is frequently used to evaluate the impact of toxicants, it is not commonly found in tropical waters; cladocerans from tropical and subtropical waterbodies should be used in bioassays. Here, we evaluated the impact of crude cyanobacteria extracts on three common, native species (*Daphnia laevis*, *Ceriodaphnia dubia*, and *Simocephalus vetulus*) based on acute and chronic bioassays. We analyzed the toxicity of cyanobacterial consortium collected from Lake Zumpango, Mexico. The FCHAB was dominated by *Planktothrix agardhii* (1.16 × 10^6^ ind mL^−1^). A series of freeze/thaw/sonification cycles at 20 kHz was used to extract the toxic metabolites and the concentration of dissolved microcystin-LR equivalents was measured using an ELISA immunological kit. *S. vetulus* was the most sensitive species, with a median lethal concentration of 0.43 compared to 1.19 µg L^−1^ of *D. magna* at 48 h. *S. vetulus* was also the most sensitive in chronic evaluations, showing a negative rate of population increase (−0.10 d^−1^) in experiments with 20% crude extract.

## 1. Introduction

Freshwater cyanobacterial harmful algal blooms (FCHABs) are characterized based on their dominance over other primary producers and abundances exceeding 10^4^ cells mL^−1^ [1,2]. Anthropogenic eutrophication, from excessive input of nitrogen and phosphorus compounds from wastewater, agriculture, and industry increases their densities [3,4]. Simultaneously, hydrological changes, nutrient-rich runoff, global warming, and elevated atmospheric carbon dioxide levels enhance their growth rates, spread, and early inoculation of surface waters [5,6]. Long-term eutrophication has been related to plankton communities restructure, reducing cladoceran biomass with cascading effects disrupting aquatic food webs [7].

Cyanobacteria have the capacity to produce bioactive metabolites with diverse structures, finding applications in the food, cosmetics, agrochemical, and pharmaceutical industries [8]. However, among these metabolites, cyanotoxins, which are produced due to genetic and environmental factors, may have an adverse effect on the health of organisms [9,10]. Cyanotoxins encompass cyclic peptides such as microcystins and nodularins, alkaloids like anatoxin-a, cylindrospermopsin, and saxitoxins, as well as non-protein amino acids like BMAA and various lipopolysaccharides [11,12]. Microcystins-LR are considered the most common toxins in freshwater systems, posing risks due to their inhibition of protein phosphatases [13]. The World Health Organization (WHO) has established guidelines of <1 μg L^−1^ for drinking water sources, or >24 μg L^−1^ for recreational waters [14]. The adverse effects of cyanotoxins can be estimated by studying the effects of purified toxins [13] or crude extracts of cyanobacterial blooms [11]. Data from the latter permits a better extrapolation of results to field conditions since blooms rarely comprise a single cyanobacterium species. Such studies also take into consideration synergistic or antagonistic effects among the secondary metabolites of various species [15].

Cladocerans are primary consumers that control phytoplankton populations and bacteria from the microbial loop, serving as a trophic link for higher-level consumers [16]. *Daphnia magna* is globally used in ecotoxicological and environmental toxicology assessments, including those evaluating zooplankton–cyanobacteria interactions. However, its application in tropical and subtropical regions is questionable because this species is restricted to Holarctic regions [17]. *D. magna* is considered an exotic species, exhibiting higher tolerance to toxicants compared to native species [18]. A recent study shows its presence in a wetland in Mexico, probably having escaped from laboratory cultures during ecotoxicological tests [19]. It is, therefore, important to conduct ecotoxicological tests using native taxa in bioassays for a more reliable extrapolation of results to natural conditions and to avoid the introduction of exotic species [15]. They also help identify any synergistic or antagonistic effects of different chemicals present in cyanobacterial blooms. However, given the diversity of metabolites produced by cyanobacteria and the complexity of cyanobacterial communities, the use of multiple toxicological approaches is necessary [20]. Therefore, research in this field, specifically through bioassays, is essential for developing effective strategies for monitoring and managing cyanobacterial blooms and their impact on aquatic ecosystems and human health [15].

We hypothesized that native cladoceran species could be more sensitive to crude extracts (unpurified mixtures from the total lysis of a cyanobacterial bloom, containing all soluble compounds, including toxins), even if they have coexisted with the cyanobacterial species. These taxa can then be used in toxicity bioassays instead of the non-native *Daphnia magna*. The aim was, therefore, to determine and compare the LC_50_ of *D. magna*, *D. laevis*, *S. vetulus*, and *C. dubia* exposed to a crude extract of cyanobacteria from Lake Zumpango (Mexico) and evaluate their tolerance to the extract compared to *D. magna*. We also assessed the ecotoxicological effect of the extract in sub-lethal concentrations throughout their life cycle for the most tolerant and the most sensitive native species and compared them with *D. magna*. 

## 2. Results

### 2.1. Cyanobacterial Abundance and Microcystin-LR Concentration in the FCHAB

Three species of cyanobacteria dominated the FCHAB, *Planktothrix agardhii*, *Cylindrospermopsis raciborskii*, and *Dolichospermum planctonicum*. *P. agardhii* dominated, accounting for 99.99% of the total biomass and an abundance of 1,166,600 ind mL^−1^. On the other hand, *D. planctonicum* exhibited the lowest abundance at 3 ind mL^−1^. The concentration of microcystin-LR equivalents in the crude extract (100%) was 10.079 µg L^−1^.

### 2.2. Acute Toxicity and Relative Sensitivity of Native Cladocerans

The LC_50_ value for *D. magna* was 11.85% (1.19 µg L^−1^). *D. leavis* showed the highest tolerance to the crude extract among the native species, with an LC_50_ value of 9.50% (0.95 µg L^−1^), whereas *C. dubia* had an LC_50_ value of 5.56%, and *S. vetulus* was the most sensitive to the crude extract, with an LC_50_ value of 4.36% (Figure 1). Among the native species, *D. laevis* had a relative sensitivity (RS) value of −0.25 at 24 h, indicating greater tolerance to the extract as compared to *D. magna*. Nevertheless, after 48 h of exposure, this value was 0.097, indicating that with time it became less tolerant than *D. magna*. *S. vetulus* was found to be the most sensitive species in both test periods (24 and 48 h), with RS values of 0.41 and 0.44, respectively (Figure 2).

### 2.3. Sublethal Effects of Native Cladocerans and D. magna

A clear dose–response relationship was observed with a sustained decrease in survival as the concentration of the crude extract and the exposure time increased (Figure 3). It was evident that the adverse effects of the accumulated cyanotoxins were more pronounced with prolonged exposure. *S. vetulus* showed similar survivorship compared to the control at 5% concentration, although high mortality occurred in the first ten days. At the highest concentration (20%), the cladocerans showed reduced survival compared to the controls, and *D. magna* was also severely affected at the higher concentration, with a 67% decrease in survival compared to controls. In contrast, *D. laevis* showed only a 2.5% diminution at the same concentration.

Crude extracts had different effects on the reproduction of the exposed cladocerans. The number of offspring per clutch increased in *D. laevis* and *D. magna* with an increasing concentration as a response to stress. However, reproduction decreased in *S. vetulus* at the highest concentrations. *S. vetulus* showed maximum reproduction in the low-toxin treatments (5 and 10% crude extract), but there was a two-day delay in the onset of reproduction compared to the control (Figure 3). In the 20% extract, this variable was reduced and delayed, with the first reproduction occurring on day 15 and lasting only for five days. *D. laevis* had its highest reproductive output on day 72 under the 20% cyanotoxin treatment, with 7.5 neonates ind^−1^ d^−1^. On the other hand, *D. magna* showed its maximum reproductive output in the treatment with the lowest toxin concentration on day 77 of the experiment. In this case, the reproductive period lasted only 26 days, 70% less than in the control (Figure 3).

*D. laevis* had a life expectancy of 51 days in the control. In the treatment with 20% extract containing cyanotoxins, life expectancy was 40 days, which was statistically lower (*p* < 0.05, Tukey test). The gross and net reproductive rates had the same pattern, demonstrating higher reproduction when the cladocerans were exposed to the crude extract. Despite this, the treatment containing 10% toxin achieved the highest values, with 147 and 80 neonates female^−1^ in the net and gross reproductive rates, respectively, both being higher than the toxin-free treatment (Figure 4). The generation time among the treatments ranged from 28 to 40 days. The population growth rate (*r*, per day) indicated that cladocerans exposed to different proportions of the crude extract compensated for the stress by reproducing earlier. The population growth rate in the treatment with 20% of the crude extract was 0.25 d^−1^, while the control had a value of 0.16 d^−1^ (see Figure 4).

The experiment with *S. vetulus* showed a significant decrease in the average lifespan as the proportion of crude extract increased, reaching 9.5 days (half of that in controls) in the treatment with 20% of crude extract. The 5% treatment exhibited the highest offspring production, exceeding 69 neonates female^−1^, compared to the control, which produced 26.60 neonates female^−1^. However, in the 20% extract treatment, there was a significant reduction to 2.20 neonates female^−1^ (*p* < 0.001, one-way ANOVA). The net reproductive rate at 5% and 10% concentration of the extract had generation times of 23.74 and 25.76 days, respectively, with no statistical differences compared to the control (*p* > 0.05, one-way ANOVA); in the 20% treatment, the generation time was 8.65 days. *S. vetulus* was more sensitie to extract concentrations of 10% and 20%, with significantly lower population growth rates compared to the control and the 5% treatment, with values of 0.07 and −0.10 d^−1^, respectively.

*D. magna* had a lifespan of ~50 days when exclusively fed on *S. acutus*. However, in treatments with cyanotoxins, the lifespan decreased in the following order: 5% (38 days), 10% (27 days), and 20% (13 days). The gross reproductive rate showed a slight, but not statistically significant (*p* > 0.05, Tukey test), increase in the 5% and 10% treatments compared to the control.). The net reproductive rate in the 20% extract treatment reduced to 14.18 neonates female^−1^, which is 66% lower than the control. The generation time decreased as the extract percentage rose; the control averaged 40 days, whereas the highest proportion (20%) resulted in approximately 16 days. Population growth rates of *D. magna* in the presence of crude extracts were 0.22 and 0.24 d^−1^ for the 5% and 10% treatments, respectively, while the 20% treatment and control showed similar rates at 0.19 d^−1^ (Figure 4).

## 3. Discussion

FCHABs are dense and persistent in the tropics, especially in systems subject to anthropogenic eutrophication. The year-round dominance of toxin-producing FCHABs in Lake Zumpango, with dissolved microcystin-LR concentrations reaching up to 11.70 µg L^−1^, poses a risk to aquatic life. Our observations indicated that *P. agardhii* was the dominant species. However, other evaluations in the same aquatic system have shown even higher concentrations of microcystins, such as 62.4 µg L^−1^, with a dominance of *Microcystis aeruginosa* [21,22]. The World Health Organization (WHO) has established a provisional limit of 10 to 20 μg L^−1^ of MC-LR for recreational water, indicating that concentrations above this threshold pose a high risk to health [23]. The concentrations evaluated in our experiments are below these levels (0.69 to 3.25 μg L^−1^ of MC-LR equivalents); however, our results suggest that even these concentrations have an adverse effect on the survival and fecundity of cladocerans.

Various mechanisms explain how crude extracts alter the biological efficiency of the species under study. Notably, dissolved microcystins alone can adversely impact cladocerans, as shown by Rohrlack et al. [24], who detailed the intoxication process of *Daphnia galeata* upon consuming *Microcystis* strains producing microcystins. Symptoms included exhaustion leading to starvation, structural alterations in intestinal epithelia, and increased microcystin concentrations in the bloodstream.

In acute evaluations, *D. laevis* was the least sensitive to the crude extract among the native species. This aligns with the findings of Nandini et al. [25] and Ferrão-Filho et al. [26], who observed low mortality and no swimming paralysis in this cladoceran when it was exposed to two cyanobacteria strains (*Microcystis aeruginosa* and *Cylindrospermopsis raciborskii*) in acute and chronic tests. This tolerance has also been demonstrated in other daphniids; for example, Pawlik-Skowrońska et al. [27] found that the 24 h LC_50_ for *Daphnia pulex* exposed to purified microcystin-LR was 3320 μg L^−1^. However, when exposed to crude extracts of *P. agardhii*, it was more sensitive, possibly due to the combination of multiple oligopeptides in the crude extracts that increase toxicity.

*S. vetulus* was the most sensitive tropical species to the crude extract, which can be attributed to its high filtration rate and non-selective feeding behavior, leading to greater cyanotoxin consumption [28]. Additionally, body size is crucial as it involves greater toxin retention in invertebrates [29,30]. *C. dubia* showed similar results to *S. vetulus* in LC_50_ tests, possibly due to smaller organisms having higher metabolic and filtration rates [31,32]. Both *D. laevis* and *D. magna* presented similar results, making them good options for ecotoxicological bioassays due to their taxonomic similarities.

In the chronic evaluations of *D. laevis* survival, no significant changes were observed between treatments. Nandini et al. [25] demonstrated that *D. laevis* can grow well on a toxic strain of *M. aeruginosa*. Additionally, Ferrão-Filho et al. [26] found this species had a high activity of the enzymes glutathione-S-transferase and catalase and promoted cellular detoxification and resistance in this cladoceran to toxic food. The tolerance of cladocerans to cyanotoxins depends on the strain of the species and its gut microbiota, as shown by Macke et al. [33].

*D. magna* and *S. vetulus*, showed high survival rates under low concentrations of crude extract; however, survival diminishes with increasing extract concentrations. Similar observations were made by Huang et al. [34], who documented decreased survival and offspring numbers in cladocerans due to reduced feeding rates attributed to toxicity. Our experiments with *D. magna* and *S. vetulus*, also show reduced fecundity in the higher concentration (20%). Similarly, Hulot et al. [35] conducted life history assays on this holarctic cladoceran exposed to extracts of *P. agardhii*, a microcystin-RR-producing, revealing accelerated life histories, altered fecundity, shortened generation times, and decreased survival rates. These results are similar to our findings, as demographic variables from life tables indicate that increased crude extract proportions lead to diminished life expectancy and longer generation times across all species. Elevated stress levels prompt organisms to adopt alternative life history strategies, investing more energy into reproduction. Nandini et al. [25] also showed that in two clones of the cladoceran *Moina macrocopa*, the clone isolated from environments with cyanobacteria exhibited earlier reproduction, increasing this variable by 15% and shorter generation times.

*D. laevis* exhibited high fecundity rates; Dao et al. [36] also reported increased fecundity but reduced survival in *D. magna* exposed to crude cyanobacterial extracts. Unfavorable environmental conditions stimulate increased fecundity, leading to substantial alterations in life expectancy [37]. Conversely, in the highest extract concentration treatment, *D. magna* achieved only a maximum fecundity of five individuals per day with elevated mortality rates. A similar outcome was observed with *S. vetulus*, which produced fewer neonates under the same conditions. This suggests that exposure to microcystins and other metabolites in cyanobacterial extracts adversely affects reproductive processes, as demonstrated by Pawlik-Skowrońska et al. [27] with *Daphnia pulex* exposed to cyanobacterial extracts containing high concentrations of microcystins, oligopeptides, anabaenopeptins, aeruginosamides, aeruginosins, and cyanopeptolins.

Choosing native species or standard bioassay organisms in ecotoxicological assessments is a decision that should be based on the specific purpose of the assessment. It is important to conduct ecotoxicological assays on native species as the use of *D. magna*, typically not from the tropics, may not have the same sensitivity as native species. Santos-Medrano and Rico-Martínez [16] have emphasized this concern, highlighting that the responses of *D. magna* might not be representative of the effects on native species. The use of local species offers the advantage of generating results that are specific and applicable to provincial ecosystems [38]. In contrast, the use of sentinel cladocerans such as *D. magna*, which has well-established laboratory protocols, offers a global perspective [39]. A key aspect that has prompted reconsideration of the organisms used in bioassays is the influence of local adaptation. Recent research has shown that, even within the same species, tolerance levels to contaminants can vary depending on the evolutionary history of populations [16,33]. This variability highlights the risk of extrapolating results obtained from non-local populations to predict ecological impacts in other aquatic systems. Ultimately, a combined approach using both established models and local taxa will be essential for advancing environmental protection and understanding eco-evolutionary processes.

The effects observed on native cladoceran species exposed to the cyanobacterial consortium dominated by microcystin-producing *P. agardhii* suggest possible alterations in the food web dynamics of the ecosystem. The pronounced sensitivity and population collapse of *S. vetulus* may lead to changes in zooplankton community structure, potentially favoring more resistant species such as *D. laevis*, which show compensatory reproductive responses even in the face of toxic stress. These changes in species composition could reduce grazing pressure on cyanobacteria, thus favoring their proliferation and hindering the transfer of carbon and polyunsaturated fatty acids to higher trophic levels [40]. The short generation time and higher fecundity of *D. laevis* make it a good option for further study on biomanipulation and the control of cyanobacterial blooms [41,42].

## 4. Conclusions

In acute tests, *S. vetulus* was the most sensitive compared to the standard cladoceran *D. magna* to the crude extract of microcystin-producing cyanobacteria. Based on the relative sensitivity obtained, it is highlighted that native species showed lower tolerance to the presence of the crude extract of cyanobacteria compared to *D. magna*. The crude extract had negative effects on all cladocerans evaluated in chronic bioassays, altering demographic variables associated with survival and fecundity. Among the native species, *D. laevis* was more tolerant to crude extracts. These suggest the need to include native cladocerans in ecotoxicological evaluations to protect local waterbodies.

## 5. Materials and Methods

### 5.1. Isolation and Maintenance of Bioassay Organisms

One parthenogenetic female from each of the three cladoceran species *D. laevis*, *C. dubia*, and *S. vetulus* was isolated from Lake Zumpango, a shallow lake (<4 m) located at a high altitude (2250 m) within the Hydrological Region of the Lacustrine Complex of the Basin of Mexico. As previously reported, this area presents eutrophication and toxic cyanobacterial dominance [22]. A population of *D. magna* was provided by the Mexican Institute of Water Technology (IMTA, Mexico, Morelos), and monoclonal cultures were established using U.S. EPA medium, and maintained under laboratory conditions over two years. All cladocerans were fed on green alga *Scenedesmus acutus*, which was grown on Bold basal medium [43].

### 5.2. Preparation of the Crude Extract of the Cyanobacterial Bloom

The cyanobacterial bloom sample was collected from the littoral zone of Lake Zumpango and placed in a 10 L plastic container. A subsample of 50 mL was preserved in 3% formalin to determine and quantify the abundances of the predominant species within the cyanobacterial consortium. In the laboratory, the samples were filtered to remove zooplankton (rotifers, copepods, or cladocerans) and were checked under a microscope (Nikon, Model: Eclipse E600, Tokyo, Japan) to guarantee that the sample contained only cyanobacteria. The sample was subjected to five cycles of freezing at −70 °C and thawing, and subsequently sonicated for 10 min at 20 KHz to ensure the cell lysis and cyanotoxins extraction [44]. Finally, the sample was centrifuged at 4000 rpm and filtered through a 0.45 µm pore-size Millipore filter to remove cell debris, including bacterial cells. The crude extract was kept frozen (−70 °C), until it was used in the bioassays with the cladocerans.

Microcystin concentration was determined using the commercial kit based on the ELISA assay “QuantiplateTMKit for Microcystins” from EnvirologixTM (Portland, Maine, USA), following the manufacturer’s specifications. Microcystin concentrations were expressed as microcystin-LR equivalents (µg L^−1^). Although the ELISA kit used is based on monoclonal antibodies directed against the ADDA fraction, it exhibits low levels of cross-reactivity with three microcystin variants (RR, YR, and LR). We interpolated the microcystin concentration in the bioassays developed. We did not evaluate other cyanotoxins; however, the extract could contain dissolved metabolites in addition to microcystins, which reflects the natural chemical complexity of the blooms and allows for a more realistic assessment of biological effects.

### 5.3. Identification and Quantification of Cyanobacteria

The preserved sample was photomicrographed using the computer application Motic images plus 3.0. Their morphological characteristics were used for species identification following standards in the literature [45]. Cyanobacterial species present in the samples were quantified by counting the entire surface of a Sedgwick–Rafter chamber at 100× magnification. For smaller or unicellular species, enumeration was performed using a Neubauer chamber at 200× magnification. Cyanobacterial density was expressed as the number of individuals per milliliter (individuals mL^−1^), calculated from the mean of three replicates.

### 5.4. Acute Toxicity Tests

Toxicity tests were conducted following the guidelines of the U.S. Environmental Protection Agency. Twenty neonates (<24 h) of each cladoceran species were subjected to six concentrations of the crude extract (ranging from 1.25, 2.5, 5, 10, 20, to 40%) and U.S. EPA as mediums. These corresponded to the concentrations of microcystin-LR equivalents of 0.125, 0.25, 0.503, 1.007, 2.015, and 4.031 µg L^−1^. The experiments were conducted in borosilicate vessels containing 40 mL plus the chosen crude extract concentration. The test cladocerans were incubated at 20 ± 2 °C without food, and experiments were conducted in quadruplicate. The survival of the individuals was recorded after 48 h using a stereo microscope. Based on the survivorship data, the lethal concentration (LC_50_) was calculated using the Probit method. The relative sensitivity (RS) was calculated using the formula established by Von der Ohe and Liess [46].

Relative sensitivity, RS = log (LC_50_ *D. magna*/LC_50i_), where LC_50_ *D. magna* represents the LC_50_ value obtained for *D. magna* exposed to a toxicant. LC_50i_ represents the LC_50_ value for species i (each native cladoceran species). A zero value indicates that the species sensitivity is equivalent to *D. magna*. A positive value indicates that *D. magna* is less sensitive, while a negative value indicates that *D. magna* is more sensitive.

### 5.5. Chronic Toxicity Test (Life Table)

Based on the data of LC_50_, we selected the most tolerant (*D. laevis*) and the most sensitive (*S. vetulus*) native species for chronic tests. Twenty neonates of each species were placed in a 40 mL volume, using three sublethal concentrations of 5, 10, and 20% of the average value of LC_50_ obtained from the four species. These concentrations also contained U.S. EPA medium as a diluent and *Scenedesmus acutus* as food (1 × 10^6^ cells mL^−1^). Each experiment included a control free of toxicant, and each treatment had four replicates. Every day, the surviving individuals were quantified and transferred to fresh test jars with appropriate medium. Neonates and dead individuals were also counted but discarded. The experiment continued until all individuals of the initial cohort died. From the data of survival and fecundity, table parameters were calculated using the formulae proposed by Krebs [47]:

Life expectancy:$$e_{x}=\frac{T_{x}}{n_{x}}$$

Gross reproductive rate:$$=\sum_{0}^{\infty }m_{x\infty }$$

Net reproductive rate:$$=\sum_{0}^{\infty }l_{x}m_{x}$$

Generation time:$$T=\frac{\sum l_{x}m_{x}x}{R_{0}}$$

Rate of population increase (*r*), solved iteratively:$$\sum_{x=w}^{n}e^{-rx}l_{x}m_{x}=1$$

### 5.6. Data Analysis

Statistical analysis of the data and graphs of both evaluations were performed with Sigma Plot 11.0 software (Systat Software, Inc., San Jose, CA, USA). Statistical analyses were performed by analysis of variance (ANOVA), followed by Tukey’s test to identify significant effects.

## Figures and Tables

**Figure 1 toxins-17-00277-f001:**
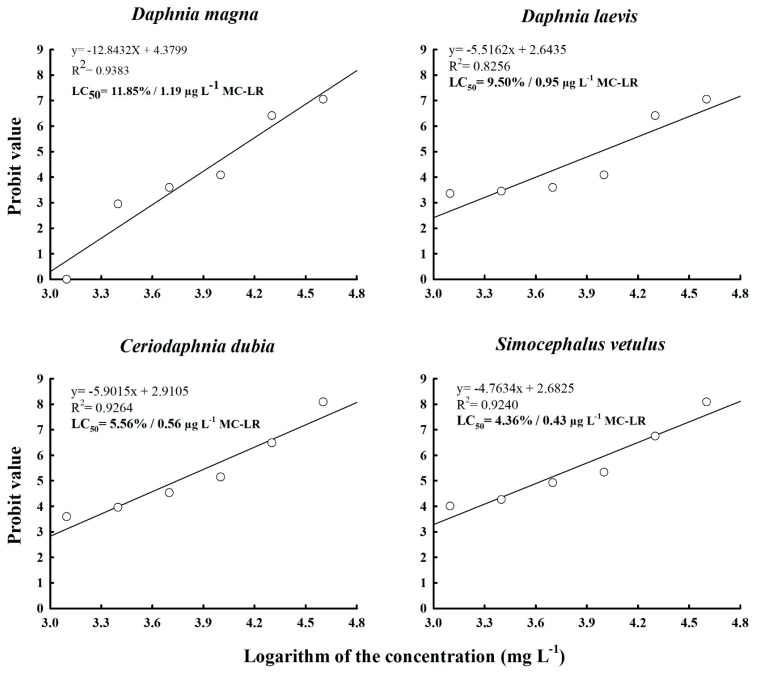
LC_50_ values of *D. magna*, *D. laevis*, *C. dubia*, and *S. vetulus* exposed to the crude extract of cyanobacteria collected from Lake Zumpango.

**Figure 2 toxins-17-00277-f002:**
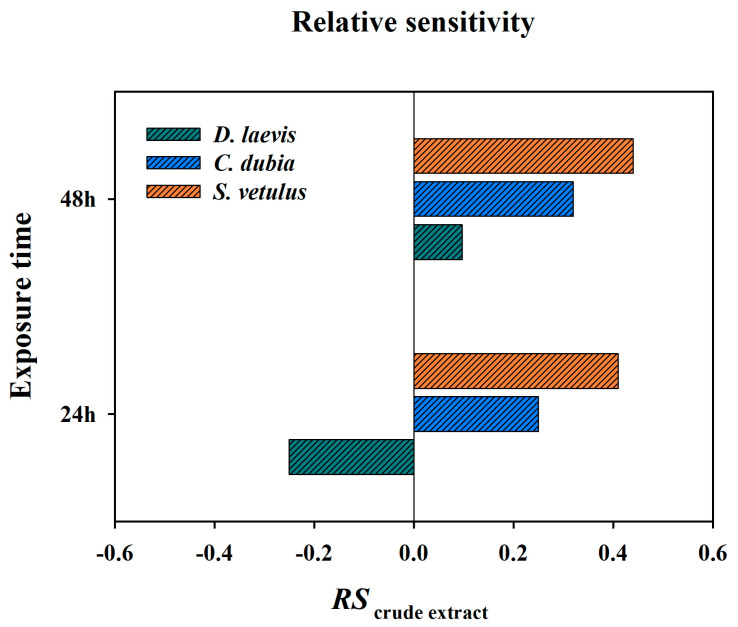
Relative sensitivity values of native species *(D. laevis*, *C. dubia*, and *S.vetulus*) when exposed to cyanobacteria crude extract during 24 and 48 h test periods.

**Figure 3 toxins-17-00277-f003:**
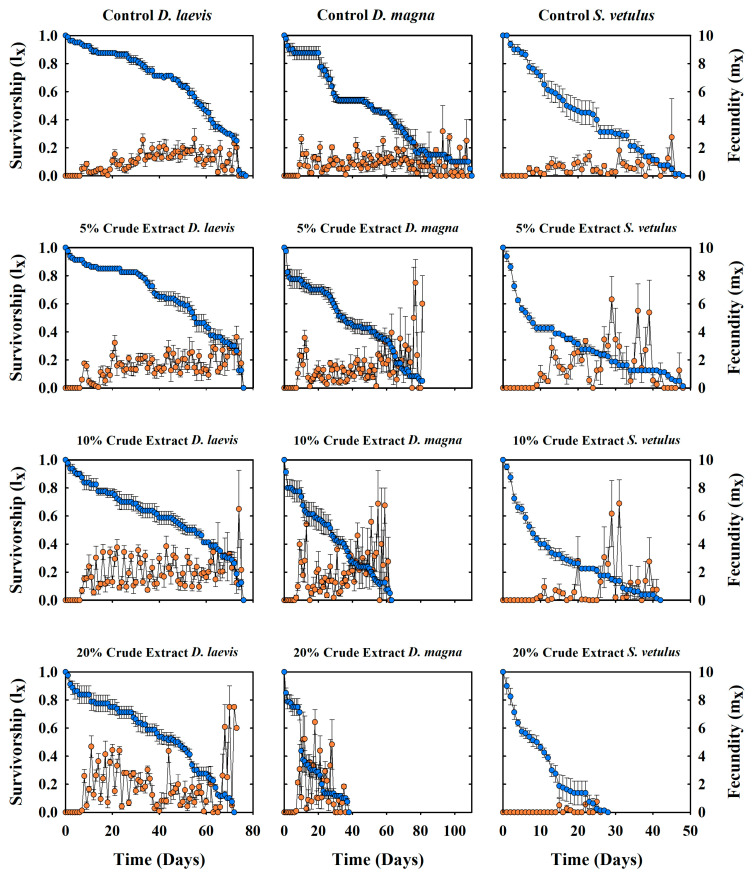
Survivorship and fecundity and patterns of *D. laevis*, *D. magna*, and *S. vetulus* populations exposed to different concentrations of cyanobacterial crude extracts (5, 10, and 20%) and a control group fed on *S. acutus*. The values represent the mean and ± standard error of four replicates (cohorts).

**Figure 4 toxins-17-00277-f004:**
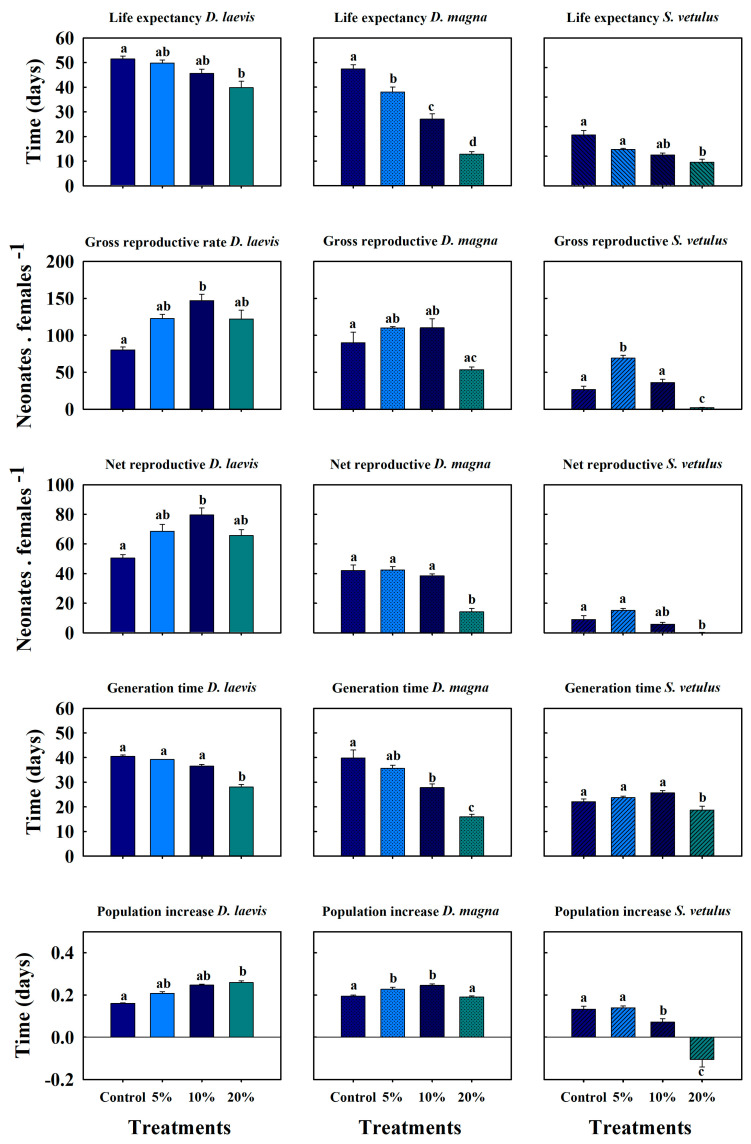
Average lifespan, gross reproductive rates, net reproductive rates, generation time, and rate of population increase of *D. laevis*, *D. magna*, and *S. vetulus* exposed to cyanobacterial crude extracts (5, 10, and 20%). Data represent mean ± standard error of four replicates. For a given variable, different alphabets indicate significant difference (*p* < 0.05, Tukey test).

## Data Availability

The original contributions presented in this study are included in the article. Further inquiries can be directed to the corresponding authors.

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
