# Peer review of "Comparative Ecotoxicological Effects of Cyanobacterial Crude Extracts on Native Tropical Cladocerans and Daphnia magna"

_toxins, 2025, doi:10.3390/toxins17060277_

Round 1
Reviewer 1 Report
Comments and Suggestions for Authors
The manuscript (MS) studied the ecotoxicological impact of freshwater cyanobacterial blooms on native cladocerans and Daphnia magna. Generally, the topic presented in this MS is interesting, and is believed to provide some useful information for related reading audiences. The MS is well prepared. But, it still need some minor improvement before its acceptance by the journal. I have listed some main points.
- The first letter of the keyword section needs to be capitalized, please be consistent.
- There are no tables in the entire text, it is recommended to add tables.
- Line 40, “Cyanobacteria have the capacity to produce bioactive metabolites with diverse structures, finding applications in the food, cosmetic, agrochemical, and pharmaceutical industries”, when referring to the cosmetics industry, the plural "cosmetics" is commonly used.
- Line 99, there is a missing space before the number "(Fig.3)".
- Line 177, There is an extra "Fig.1" in the title of Figure 1, please delete it.
- Figure 2. statistical data should be included in the figures.
Author Response
Answers to Reviewer 1
We thank the reviewer for helpful comments. We have complied with as far as possible.
Reviewer 1
The manuscript (MS) studied the ecotoxicological impact of freshwater cyanobacterial blooms on native cladocerans and Daphnia magna. Generally, the topic presented in this MS is interesting, and is believed to provide some useful information for related reading audiences. The MS is well prepared. But, it still need some minor improvement before its acceptance by the journal. I have listed some main points.
Comment: The first letter of the keyword section needs to be capitalized, please be consistent.
Reply: Thank you for your observation, we have corrected it.
Comment: There are no tables in the entire text, it is recommended to add tables.
Reply: The data presented as figures is helpful to know the trends than a table.
Comment: Line 40, “Cyanobacteria have the capacity to produce bioactive metabolites with diverse structures, finding applications in the food, cosmetic, agrochemical, and pharmaceutical industries”, when referring to the cosmetics industry, the plural "cosmetics" is commonly used.
Reply: Corrected
Comment: Line 99, there is a missing space before the number "(Fig.3)".
Reply: We have now added the space
Comment: Line 177, There is an extra "Fig.1" in the title of Figure 1, please delete it.
Reply: Deleted
Comment: Figure 2. statistical data should be included in the figures.
Reply: It is not feasible to include a statistic in this context, as the information is derived from a single unit value obtained from a LC50 test.

Reviewer 2 Report
Comments and Suggestions for Authors
The study addresses an important issue in ecotoxicology, especially in tropical freshwater systems where standardized species like Daphnia magna may not reflect local ecological responses. The inclusion of native cladocerans—Daphnia laevis, Ceriodaphnia dubia, and Simocephalus vetulus—adds ecological relevance and strengthens the applicability of the findings for regional environmental assessment.
The experimental approach is robust, combining acute and chronic bioassays and life table analyses to evaluate species-specific sensitivity to crude cyanobacterial extracts. The study provides compelling evidence that native species may respond more severely to local cyanotoxins than non-native model species, reinforcing the value of using indigenous organisms in ecotoxicological evaluations.
That said, I recommend a revision of the manuscript. First, the title should be improved for clarity and specificity. I suggest the following revised title:
"Comparative Ecotoxicological Effects of Cyanobacterial Crude Extracts on Native Tropical Cladocerans and Daphnia magna"
This version more accurately reflects the comparative design of the study, the use of crude extracts, and the tropical context of the native species tested.
The manuscript also requires thorough language editing. Several grammatical errors and awkward phrases should be corrected. For example, the abstract includes the phrase “S. vetulus also turned out to be the lowest sensitivity,” which should be revised to “S. vetulus showed the lowest tolerance in chronic evaluations.” The phrase “a Daphnia magna” should also be corrected throughout. Terms like “diminution” should be replaced with clearer alternatives such as “decrease.” A complete language review is recommended to ensure clarity and consistency.
Species names should follow standard taxonomic conventions: write the full name on first use, then use the abbreviated form (e.g., D. magna) throughout the remainder of the text. This should be applied consistently for all species mentioned in the manuscript.
Figures 3 and 4 require improvements in formatting. The font size for axis labels should be increased, color choices should accommodate color-blind readers, and figure legends should be more detailed to allow standalone interpretation. Residual placeholder text from the manuscript template (e.g., “this section may be divided by subheadings”) should also be removed.
In the methods, it would be helpful if the authors clarified whether the crude extract was assessed for potential bacterial contamination or other co-occurring bioactive metabolites. Additionally, a brief note on the limitations of ELISA (e.g., matrix effects or potential cross-reactivity) would be useful for readers interpreting the microcystin quantification.
The discussion is generally well developed, but the ecological context could be deepened. I encourage the authors to briefly address how observed effects on zooplankton life history traits could influence broader food web dynamics, primary productivity, or nutrient cycling in eutrophic tropical lakes. This would enhance the environmental relevance of the findings.
Lastly, the authors should incorporate the references suggested in the manuscript comments into the Introduction section. These references provide useful context on cyanobacteria-zooplankton interactions, eutrophication, ecological risk, and species sensitivity, and would help frame the study within the broader literature. The following references are particularly relevant:
-
https://doi.org/10.1007/s10021-024-00908-x
-
https://doi.org/10.3390/phycology4010010
Once revised, the study will make a meaningful contribution to the literature on tropical freshwater ecotoxicology.
Comments on the Quality of English LanguageSeveral grammatical errors and awkward phrases should be corrected, a complete language review is recommended to ensure clarity and consistency.
Author Response
Answers to Reviewer 2
We are grateful to the Reviewer 2 for improving our presentation. We have gratefully taken into account of all the suggestions and modified our manuscript accordingly.
Reviewer 2
The study addresses an important issue in ecotoxicology, especially in tropical freshwater systems where standardized species like Daphnia magna may not reflect local ecological responses. The inclusion of native cladocerans—Daphnia laevis, Ceriodaphnia dubia, and Simocephalus vetulus—adds ecological relevance and strengthens the applicability of the findings for regional environmental assessment.
The experimental approach is robust, combining acute and chronic bioassays and life table analyses to evaluate species-specific sensitivity to crude cyanobacterial extracts. The study provides compelling evidence that native species may respond more severely to local cyanotoxins than non-native model species, reinforcing the value of using indigenous organisms in ecotoxicological evaluations.
Comment: That said, I recommend a revision of the manuscript. First, the title should be improved for clarity and specificity. I suggest the following revised title: "Comparative Ecotoxicological Effects of Cyanobacterial Crude Extracts on Native Tropical Cladocerans and Daphnia magna"This version more accurately reflects the comparative design of the study, the use of crude extracts, and the tropical context of the native species tested.
Reply: We appreciate your comment and have considered it. We have changed the title.
Comment: The manuscript also requires thorough language editing. Several grammatical errors and awkward phrases should be corrected. For example, the abstract includes the phrase “S. vetulus also turned out to be the lowest sensitivity,” which should be revised to “S. vetulus showed the lowest tolerance in chronic evaluations.” The phrase “a Daphnia magna” should also be corrected throughout. Terms like “diminution” should be replaced with clearer alternatives such as “decrease.” A complete language review is recommended to ensure clarity and consistency.
Reply: Thank you for the comment. We have carefully reviewed the entire manuscript to address grammatical issues and to improve clarity.
Comment: Species names should follow standard taxonomic conventions: write the full name on first use, then use the abbreviated form (e.g., D. magna) throughout the remainder of the text. This should be applied consistently for all species mentioned in the manuscript.
Reply: Thank you for your observation. We have complied with the suggestion
Comment: Figures 3 and 4 require improvements in formatting. The font size for axis labels should be increased, color choices should accommodate color-blind readers, and figure legends should be more detailed to allow standalone interpretation. Residual placeholder text from the manuscript template (e.g., “this section may be divided by subheadings”) should also be removed.
Reply: We appreciate your comments. We have improved the formatting of Figures 3 and 4 by increasing the font size of the axis labels, selecting colorblind-friendly palettes (corroborated with a special filter).
Comment: In the methods, it would be helpful if the authors clarified whether the crude extract was assessed for potential bacterial contamination or other co-occurring bioactive metabolites. Additionally, a brief note on the limitations of ELISA (e.g., matrix effects or potential cross-reactivity) would be useful for readers interpreting the microcystin quantification.
Reply: We have added information to clarify these points in the Methods section.
Comment: The discussion is generally well developed, but the ecological context could be deepened. I encourage the authors to briefly address how observed effects on zooplankton life history traits could influence broader food web dynamics, primary productivity, or nutrient cycling in eutrophic tropical lakes. This would enhance the environmental relevance of the findings.
Lastly, the authors should incorporate the references suggested in the manuscript comments into the Introduction section. These references provide useful context on cyanobacteria-zooplankton interactions, eutrophication, ecological risk, and species sensitivity, and would help frame the study within the broader literature. The following references are particularly relevant:
- https://doi.org/10.1007/s10021-024-00908-x
- https://doi.org/10.3390/phycology4010010
Once revised, the study will make a meaningful contribution to the literature on tropical freshwater ecotoxicology.
Reply: Thank you for your valuable feedback. We have added a new paragraph at the end of the Discussion section to clearly address the potential implications of our findings for food web dynamics. Furthermore, we have included the reference by King et al. in the Introduction. Regarding Bisinicu et al., we believe that the environmental characteristics described in their study differ significantly from those of our shallow, eutrophic lakes, that is is why we chose not to include it.
Comment: Several grammatical errors and awkward phrases should be corrected; a complete language review is recommended to ensure clarity and consistency.
Reply: We have carefully reviewed the entire manuscript to address grammatical issues and improve clarit

Round 2
Reviewer 2 Report
Comments and Suggestions for Authors
I have carefully reviewed the revised manuscript and the authors’ responses. They have taken into consideration the comments provided in my first-round review. I have updated the second-round review report accordingly, including a few suggestions where I believe further improvement could enhance the manuscript. That said, I believe a few areas still require further refinement to enhance the clarity and overall quality of the manuscript.
Specifically, the Abstract remains overly dense and would benefit from being streamlined to more clearly and concisely present the rationale, methods, key findings, and implications.
The Introduction, while informative, could be more focused to better emphasize the knowledge gap, explicitly state the hypothesis—particularly concerning the potentially higher sensitivity of native species—and clearly define the study objectives.
The Results section is comprehensive but could be more concise in places.
Similarly, the Discussion contains some repetitive elements and would be strengthened by a tighter focus, particularly in connecting the findings to broader ecological implications such as zooplankton community structure, food web dynamics, and the management of eutrophic freshwater systems.
Author Response
We sincerely thank Reviewer for the constructive feedback. All suggestions have been carefully considered and incorporated into the revised manuscript
Comments and Suggestions for Authors
I have carefully reviewed the revised manuscript and the authors’ responses. They have taken into consideration the comments provided in my first-round review. I have updated the second-round review report accordingly, including a few suggestions where I believe further improvement could enhance the manuscript. That said, I believe a few areas still require further refinement to enhance the clarity and overall quality of the manuscript.
Comment: Specifically, the Abstract remains overly dense and would benefit from being streamlined to more clearly and concisely present the rationale, methods, key findings, and implications.
Reply: We have condensed the section by deleting and rewriting several phrases. The number of words has now been reduced from 206 to 178.
Comment: The Introduction, while informative, could be more focused to better emphasize the knowledge gap, explicitly state the hypothesis—particularly concerning the potentially higher sensitivity of native species—and clearly define the study objectives.
Reply: Thank you. We have added a paragraph and modified the study objectives and hypothesis in the last paragraph of this section.
Comment: The Results section is comprehensive but could be more concise in places.
Reply: Based on your suggestion we have deleted some statements to make this section more concise.
Comment: Similarly, the Discussion contains some repetitive elements and would be strengthened by a tighter focus, particularly in connecting the findings to broader ecological implications such as zooplankton community structure, food web dynamics, and the management of eutrophic freshwater systems.
Reply: We have restructured the section to the best of our ability. Our emphasis in this MS was to emphasize the use of native taxa in ecotoxicological assays, hence we hesitate to include points on zooplankton community structure and food webs. We have added information on the possible use of Daphnia laevis in biomanipulation and the control of cyanobacterial blooms.

Round 3
Reviewer 2 Report
Comments and Suggestions for Authors
The authors have addressed all the suggestions and comments raised in the previous round of review.